# Cell Therapy Transplant Canada (CTTC) Consensus-Based Guideline 2024 for Management and Treatment of Chronic Graft-Versus-Host Disease and Future Directions for Development

Dennis Dong Hwan Kim [1,*], Gizelle Popradi [2], Kylie Lepic [3], Kristjan Paulson [4], David Allan [5], Ram Vasudevan Nampoothiri [5], Sylvie Lachance [6], Uday Deotare [7], Jennifer White [8], Mohamed Elemary [9], Kareem Jamani [10], Christina Fraga [11], Christopher Lemieux [12], Igor Novitzky-Basso [1], Arjun Datt Law [1], Rajat Kumar [1], Irwin Walker [3] and Kirk R. Schultz [13] on behalf of the CTTC Chronic GVHD Guideline Working Group

1 Princess Margaret Cancer Centre, University Health Network, University of Toronto, Toronto, ON M5G 2M9, Canada; arjun.law@uhn.ca (A.D.L.)
2 Health Centre, McGill University, Montreal, QC H4P 2P5, Canada
3 Juravinski Cancer Centre, McMaster University, Hamilton, ON L8V 5C2, Canada
4 CancerCare Manitoba, University of Manitoba, Winnipeg, Manitoba, MB R3E 0V9, Canada
5 Ottawa Hospital Research Institute, University of Ottawa, Ottawa, ON K1H 8L6, Canada
6 Hôpital Maisonneuve-Rosemont, Department of Medicine, University of Montreal, Montreal, QC H3C 3J7, Canada; silvy.lachance@umontreal.ca
7 London Health Sciences Centre, University of Western Ontario, London, ON N6A 5A5, Canada
8 Vancouver General Hospital, British Columbia Cancer Agency, University of British Columbia, Vancouver, BC V5Z 1M9, Canada
9 Saskatchewan Cancer Agency, University of Saskatchewan, Saskatchewan, SK S7N 4H4, Canada
10 Tom Baker Cancer Centre, University of Calgary, Calgary, AB T2N 4N2, Canada; kareem.jamani@ahs.ca
11 Queen Elizabeth II Health Sciences Centre, Dalhousie University, Halifax, NS B3H 4R2, Canada
12 CHU de Québec—Université Laval, Université Laval, Québec, QC G1R 2J6, Canada
13 British Columbia Children's Hospital, University of British Columbia, Vancouver, BC V6H 3N1, Canada
* Correspondence: dr.dennis.kim@uhn.ca; Tel.: +1-(416)-946-4501x2464; Fax: +1-(416)-946-4563

**Abstract:** This is a consensus-based Canadian guideline whose primary purpose is to standardize and facilitate the management of chronic graft-versus-host disease (cGvHD) across the country. Creating uniform healthcare guidance in Canada is a challenge for a number of reasons including the differences in healthcare authority structure, funding and access to healthcare resources between provinces and territories, as well as the geographic size. These differences can lead to variable and unequal access to effective therapies for GvHD. This document will provide comprehensive and practical guidance that can be applied across Canada by healthcare professionals caring for patients with cGvHD. Hopefully, this guideline, based on input from GvHD treaters across the country, will aid in standardizing cGvHD care and facilitate access to much-needed novel therapies. This consensus paper aims to discuss the optimal approach to the initial assessment of cGvHD, review the severity scoring and global grading system, discuss systemic and topical treatments, as well as supportive therapies, and propose a therapeutic algorithm for frontline and subsequent lines of cGvHD treatment in adults and pediatric patients. Finally, we will make suggestions about the future direction of cGvHD treatment development such as (1) a mode-of-action-based cGvHD drug selection, according to the pathogenesis of cGvHD, (2) a combination strategy with the introduction of newer targeted drugs, (3) a steroid-free regimen, particularly for front line therapy for cGvHD treatment, and (4) a pre-emptive approach which can prevent the progression of cGvHD in high-risk patients destined to develop severe and highly morbid forms of cGvHD.

**Keywords:** chronic GVHD; allogeneic hematopoietic stem cell transplantation; recommendation

## 1. Introduction

Chronic graft-versus-host disease (cGvHD) is a syndrome with diverse clinical features resembling autoimmune and immunological disorders, occurring after allogeneic hematopoietic cell transplantation (HCT) [1,2]. It can affect the long-term outcomes of allogeneic HCT patients by increasing morbidity and mortality [3] and is associated with a reduced quality of life [4–6]. It often requires long-term immunosuppressive therapy, which can lead to the development of significant side effects and toxicities [1–3]. While there is no national guideline for the treatment of cGvHD in Canada [7], transplant program specific standardized protocols with similarities across the country do exist in each institution. These protocols include many systemic therapeutic options and are not just limited to systemic corticosteroids, extracorporeal photopheresis [8,9], rituximab, sirolimus, or mycophenolate mofetil [7,10,11]. Recently, newer agents have become accessible to some provinces and territories including ibrutinib [12], ruxolitinib [13], and belumosudil [14], while axatilimab is awaiting Health Canada approval [15]. However, standardized and systemic approaches are still scarce, thus demanding a nationwide standardized guideline.

Although comprehensive international guidelines on cGvHD management have been published [16–18], a Canadian-specific guideline is still necessary. There are barriers to prevent the standardization of clinical practice such as Canada's unique healthcare structure, variable funding mechanisms across the country, differences in provincial medication approval processes, the geographic size of Canada, and the patient populations that exist within the Canadian healthcare system. Accordingly, tailored recommendations are needed to optimize the management of cGvHD in Canada. Particularly, accessibility to newer agents is problematic for countries with a publicly funded healthcare system, where healthcare service models are quite different from the US-based healthcare system. This raises the issue of how to access the best treatment in countries with a restricted healthcare system like Canada, Australia, New Zealand, and other European countries, requiring innovation to manage this challenging patient population.

Through a national consensus guideline, ensuring consistency in the diagnosis and management of cGvHD across the provinces and institutions with different funding and approval systems will eventually lead to facilitating the transplant program's access to novel therapies and newer agents and will offer our patients standardized care, leading to improved patient outcomes and cost savings in the healthcare system. Accordingly, this document will serve as a consensus-based Canadian guideline for the clinical management of cGvHD patients focusing on (1) the initial assessment and diagnosis of cGvHD, (2) organ severity scoring and global grading, (3) the therapeutic approach, including systemic, topical and supportive treatment, and (4) the management of specific cGvHD patient subpopulations such as pediatric patients [19]. Finally, this paper will highlight the need for innovative approaches for the management of cGvHD with commentary on the future direction of chronic GvHD treatment development [20–24].

## 2. Consensus Process of the Cell Therapy Transplant Canada (CTTC) Guideline for Chronic GvHD Management

Current clinical practice in cGvHD management is quite heterogeneous in Canada [7]. Institutions from the various provinces and territories have similar but different practices and policies for the management of cGvHD. Accordingly, experts across Canada reached a consensus and came together to develop this Canadian consensus guideline for cGvHD management.

Initially, a first draft was established within the Princess Margaret Cancer Centre allogeneic hematopoietic stem cell transplant program in early 2023, which was based on internal consensus. Consultation was then expanded to include a broader Canadian perspective under the Cell Therapy Transplant Canada (CTTC) organization, with contributing members from different academic institutions across Canada. All members of the CTTC writing committee have shared their clinical experience and feedback. All the writing committee members have read, reviewed, and approved the final draft.

## 3. Summary of the Biology and Pathogenesis of Chronic GvHD

cGvHD occurs as a result of an immune response mounted by transplanted donor immune cells against the recipient's tissues [1,2]. In order to discuss the contemporary concepts of cGvHD treatment, it is necessary to review the pathogenesis of cGvHD. The pathophysiology of cGVHD involves a complex interplay between the donor immune cells and the recipient tissues, which leads to the deregulation of immune pathways, chronic inflammation, tissue damage, and fibrosis. The process begins with the activation of donor T cells by recipient antigens, which then proliferate and differentiate into effector T cells and attack recipient tissues, initiating an inflammatory response [10]. Also, deregulated immune pathways contribute to chronic immune activation by recipient antigens and subsequent chronic inflammation [10].

The pathogenesis of cGVHD has been characterized as arising in three phases, which occur simultaneously. The inflammatory phase includes the release of cytokines and chemokines, which recruit additional immune cells to the site of injury [10]. In the immune deregulation phase, these immune cells then amplify the inflammatory response and cause tissue damage, leading to the release of additional danger signals that further stimulate the immune response. Finally, in the fibrotic phase, chronic inflammation can lead to irreversible tissue fibrosis [10], which can impair organ function and cause long-term damage. The development of fibrosis is thought to be due to the activation of fibroblasts and myofibroblasts by the chronic inflammatory response. A fourth phase or component is suggested by the absence of regulatory populations including regulatory T cells, B cells, NK cells, and macrophages [25]. Understanding the different involvement of disease phases is fundamental for therapeutic drug selection with different action mechanisms.

## 4. The Diagnosis of Chronic GvHD

The clinical manifestations of cGVHD are very diverse and heterogeneous. cGVHD generally involves several organs or sites and is rarely restricted to a single organ. It is characterized by features that differ from the typical dermatitis, enteritis, and cholestatic liver manifestations of acute GvHD. In order to reduce the risk of missing the recognition of such symptoms/signs, identifying key features from a systematic review of the patient in the clinic should be the first step [1]. Supplementary Table S1 summarizes the organs frequently involved and the questions that can be asked in the clinic to capture the early symptoms of cGVHD [11].

The diagnosis of cGVHD is based on a combination of clinical features and histopathology, functional, and laboratory testing. According to the NIH 2014 criteria [26], the diagnosis of cGVHD requires the presence of at least one diagnostic clinical sign or feature of cGVHD, such as poikiloderma or esophageal web, lichenoid oral mucosal lesion, or sclerodermatous skin lesion, or the presence of at least one distinctive manifestation confirmed by biopsy or testing, such as keratoconjunctivitis sicca in the same or another organ (Table 1). In addition, other possible diagnoses for clinical symptoms must be excluded. No time threshold, such as day 100, is set for the diagnosis of cGVHD. cGVHD can be classified into (1) classic cGVHD (i.e., without features or characteristics of acute GvHD) or (2) an overlap syndrome in which diagnostic or distinctive features of chronic GvHD and acute GvHD appear together.

**Table 1.** Diagnostic, distinctive, and non-specific features of chronic GvHD, reprinted with permission from Ref. [26], 2024 Elsevier.

| Organ or Site | Diagnostic (Sufficient for Diagnosis) | Distinctive (Insufficient Alone for Diagnosis) | Other | Features Seen in Both Acute and Chronic GvHD |
|---|---|---|---|---|
| Skin | Poikiloderma | Depigmentation | | Erythema |
| | Lichen planus-like | | | Maculopapular |
| | Sclerosis | Papulosquamous | | Pruritus |
| | Morphea-like | | | |
| | Lichen sclerosis-like | | | |

**Table 1.** *Cont.*

| Organ or Site | Diagnostic (Sufficient for Diagnosis) | Distinctive (Insufficient Alone for Diagnosis) | Other | Features Seen in Both Acute and Chronic GvHD |
|---|---|---|---|---|
| Nails | | Dystrophy | | |
| | | Onycholysis | | |
| | | Nail loss | | |
| | | Pterygium unguis | | |
| Scalp and body hair | | Alopecia (scarring or non-scarring) | | |
| | | Scaling | | |
| Mouth | Lichen planus-like | Xerostomia | | Gingivitis |
| | | Mucoceles | | Mucositis |
| | | Mucosal atrophy | | Erythema |
| | | Pseudomembranes or ulcers | | Pain |
| Eyes | | New dry, gritty, or painful eyes (sicca) | | |
| | | Keratoconjunctivitis sicca | | |
| | | Punctate keratopathy | | |
| Genitalia | Lichen planus-like | Erosions | | |
| | Lichen sclerosis-like | Fissures | | |
| | Female: | Ulcers | | |
| | Vaginal scarring or stenosis | | | |
| | Clitoral or labial agglutination | | | |
| | Male: | | | |
| | Phimosis | | | |
| | Urethral scarring or stenosis | | | |
| GI tract | Esophageal web | | | Diarrhea |
| | Esophageal stricture | | | Anorexia |
| | | | | Nausea or emesis |
| | | | | Failure to thrive |
| | | | | Weight loss |
| Liver | | | | Total bilirubin, alkaline phosphatase or ALT > 2× upper normal limit |
| Lung | Bronchiolitis obliterans diagnosed by biopsy | | Cryptogenic organizing pneumonia | |
| | | | Restrictive lung disease | |
| Muscles, fascia, joints | Fasciitis | Myositis | | |
| | Joint stiffness or contractures due to sclerosis | Polymyositis | | |
| Hematopoietic and Immune | | Thrombocytopenia | | |
| | | Eosinophilia | | |
| | | Hypo- or hypergammaglobulinemia | | |
| | | Autoantibodies | | |
| | | Raynaud phenomenon | | |
| Others | | Pleural or pericardial effusions | | |
| | | Nephrotic syndrome | | |
| | | Myasthenia gravis | | |
| | | Peripheral neuropathy | | |

Adopted from *Blood* **2015**, *125*, 606–615 [11].

The diagnosis of cGvHD can be supported by histopathologic findings, such as the presence of lymphocytic infiltrates or collagen deposit/fibrosis in the affected tissues. However, histopathologic findings alone are not sufficient for the diagnosis of cGvHD. In cases where the "distinctive" manifestation is the only sign of cGvHD, additional confirmation is required for the diagnosis of cGvHD, such as biopsy, imaging, or a pulmonary function test (PFT).

Whenever cGvHD is suspected, pulmonary function tests (PFTs) are strongly recommended given the frequency of the asymptomatic involvement and the importance of the early recognition of lung GvHD. The bronchodilator response should be assessed to rule out asthma. In addition to objective measures, patient-reported outcome measures, such as the modified Lee Symptom Scale [27], can be used at the time of the initial diagnosis of cGvHD, particularly in the context of research purposes. Genital manifestations of cGvHD are often overlooked. Accordingly, a directed questionnaire searching for gynecological manifestations is strongly suggested. Gynecology assessment around day 100 will aid in the detection of genital GvHD in female patients, particularly in post-menopausal women where features of atrophic vaginitis may mask early cGvHD symptoms. Immunoglobulin quantitation around day 100 will help detect hypergammaglobulinemia, which is evident when B cell deregulation is the main pathway for cGvHD development, or hypogammaglobulinemia, which can predispose the patient to opportunistic infection during cGvHD treatment.

The diagnosis of cGvHD in children remains challenging. This is especially true in the diagnosis of lung cGvHD due to the fact that children tend to have more potential insults to their lungs with recurrent "childhood" respiratory virus infections, and the inability to perform standard spirometry-based pulmonary function tests before age six [28]. Alternatives include the multiple breath washout test that can be carried out, as this test is allowed down to the age of three [29]. Also, young children may not volunteer expressing symptoms such as dry eyes, and it takes an astute clinician to pick up xeropthalmia and usually more frequent exams by an ophthalmologist are required. Since most children and adolescents are not sexually active, vaginal involvement in girls is usually not recognized. Atypical manifestations of cGvHD may be more frequent in children, although this is still to be conclusively determined [30].

In summary, we recommend the use of the NIH 2014 criteria for the diagnosis of cGvHD, with a strong suggestion to include investigations in other organs not showing any symptoms/signs of cGvHD such as the lungs or genitalia.

## 5. Atypical GvHD

The NIH consensus project of 2020 emphasized atypical GvHD. This is an emerging disease entity that reflects the recently recognized features of cGvHD that differ from the classical manifestations of cGvHD (Table 2) [22,31]. It frequently involves the central nervous system, peripheral nervous system, lungs, serositis, kidney, musculoskeletal system, and immune-mediated cytopenia. Atypical cGvHD affects a substantial number of patients, and it can manifest before or without NIH-defined cGvHD features. Several risk factors for atypical GvHD were proposed such as prior acute GvHD, total body irradiation, and donor lymphocyte infusion [31].

**Table 2.** Spectrum of atypical manifestations of chronic GvHD.

| **1. Immune-mediated cytopenias** | **6. Peripheral Nervous System** |
| --- | --- |
| Immune-mediated neutropenia | Chronic inflammatory demyelinating polyneuropathy |
| Hemolytic anemia | Guillain–Barre syndrome |
| Immune-mediated thrombocytopenia | Small fiber polyneuropathy |
| Evans syndrome | Myasthenia gravis |
| Thrombotic microangiopathy | Other peripheral neuropathies |
| **2. Gastrointestinal** | **7. Renal** |
| Immune-mediated pancreatitis | Macroalbuminuria or nephrotic range proteinuria |
| **3. Pulmonary** | Glomerulonephritis and tubulointerstitial damage |
| Organizing pneumonia | Renal thrombotic microangiopathy |

**Table 2.** *Cont.*

| Non-specific interstitial pneumonia | **8. Muscles, fascia, joints** |
|---|---|
| Pleuroparenchymal pulmonary fibroelastosis | Edema |
| **4. Endocrine** | Muscle cramps |
| Thyroiditis—Hashimoto's disease | Arthralgia |
| Thyroiditis—Graves' disease | Arthritis |
| **5. Central nervous system (CNS)** | Myositis |
| Neurocognitive deficits | **9. Others** |
| Meningoencephalitis | Cardiac conduction abnormalities |
| Multiple sclerosis-like encephalitis | Cardiomyopathy/myocarditis |
| CNS vasculitis-like disorders | Vasculitis |
| | Serositis—pericardial and pleural effusions, ascites |
| | Raynaud's phenomenon |

Adopted and modified from Transplant and Cellular Therapy 28(2022) 426–445 [22].

Clinical suspicion of atypical GvHD is the first step to establish its diagnosis and management. Its pathophysiology is not fully elucidated and strongly warrants further investigation. Provisional diagnostic criteria for suspected atypical manifestations of cGvHD have been published and require further validation. Atypical GvHD manifestations frequently require a different therapeutic approach [22]. Thus, we recommend that patients with newly diagnosed cGvHD should be screened and monitored for the features of atypical cGVHD.

## 6. Organ Severity Scoring and Global Grading of Chronic GvHD

The NIH 2014 criteria provide guidelines for the organ severity scoring and global grading of cGvHD, which are based on the severity and extent of organ involvement [26]. Organs that are commonly involved in cGvHD include the skin, liver, gastrointestinal (GI) tract, oral mucosa, eyes, and lungs. Particularly for sclerotic GvHD, assessment of the photographic range of motion (P-ROM) at the time of initial diagnosis is mandatory for subsequent response assessment. For each organ, specific clinical and laboratory criteria are used to assign a score ranging from 0 to 3, where 0 indicates no involvement and 3 indicates severe involvement (Table 3).

**Table 3.** Summary of Chronic GvHD severity score in each organ, reprinted with permission from Ref. [26], 2024 Elsevier.

| Organ | Score 0 | Score 1 | Score 2 | Score 3 |
|---|---|---|---|---|
| Performance status | ECOG 0/KPS 100% | ECOG 1/KPS 80–90% | ECOG 2/KPS 60–70% | ECOG 3–4/KPS < 60% |
| Skin-chronic GvHD features | No BSA involved | 1–18% BSA | 19–50% BSA | >50% BSA |
| Skin-sclerotic features | No sclerotic involvement | | Superficial sclerotic involvement | Deep sclerotic involvement |
| Mouth | No symptoms | Mild symptoms/signs | Moderate symptoms/signs with partial limitation of oral intake | Severe symptoms/signs with major limitation of oral intake |
| Eyes | No symptoms | Mild dry eye symptoms, not affecting ADL | Moderate dry eye symptoms, partially affecting ADL | Severe dry eye symptoms, significantly affecting ADL OR unable to work OR loss of vision due to KCS |
| GI tract | No symptoms | Symptoms without weight loss (<5%) | Symptoms with mild–moderate weight loss (5–15%) OR moderate diarrhea | Symptoms with significant weight loss (>15%), requiring nutritional support OR esophageal dilatation OR severe diarrhea |

**Table 3.** *Cont.*

| Organ | Score 0 | Score 1 | Score 2 | Score 3 |
|---|---|---|---|---|
| Liver | Normal total bilirubin and ALT/AP < 3×ULN | Normal total bilirubin with ALT ≥ 3 to 5×ULN or AP ≥ 3×ULN | Elevated total bilirubin but ≤3 mg/dL or ALT > 5×ULN | Elevated bilirubin > 3 mg/dL |
| Lung | No symptoms | Mild symptoms (SOB after climbing one flight of steps) | Moderate symptoms (SOB after walking on flat ground) | Severe symptoms (SOB at rest) |
| Lung score | FEV1 ≥ 80% | FEV1 60–79% | FEV1 40–59% | FEV1 ≤ 39% |
| Joint and fascia | No symptoms | Mild tightness of arms/legs, normal or mild ROM AND not affecting ADL | Tightness of arms/legs OR joint contractures, moderate decrease in ROM AND mild to moderate limitation of ADL | Contractures with significant decrease in ROM AND significant limitation of ADL |
| Genital tract | No signs | Mild signs | Moderate signs, may have symptoms with discomfort on exam | Severe signs with or without symptoms |
| Other organ involvement *** | No functional impact | Mild impact on function | Moderate impact on function | Severe impact on function |

Adopted from Biol Blood Marrow Transplant **2015**, *21*, 389–401 [26]. Abbreviations: KPS, Karnofsky performance status; BSA, body surface area; ADL, activity of daily living; ALT, alanine transferase; AP, alkaline phosphatase; ULN, upper limit of normal; SOB, shortness of breath; FEV1, forced expiratory volume in 1 s; ROM, range of motion. *** Other organ manifestations include ascites, pericardial effusion, pleural effusion, nephrotic syndrome, myasthenia gravis, neuropathy, polymyositis, weight loss with GI symptom, eosinophilia, thrombocytopenia, etc.

The global grading of cGvHD is based on the overall severity of the disease, as determined by the organ scoring and clinical assessment [26]. The global grading system includes three categories: mild, moderate, and severe grade. The classification is based on the maximal organ severity score, the number of organs involved, and the impact of organ involvement on the patient's quality of life and daily activity level (Table 4). This allows for a more comprehensive assessment of the disease burden and facilitates the monitoring of disease progression and treatment response.

**Table 4.** Global grading of chronic GvHD.

| No. of Organs Involved | Mild Grade | Moderate Grade | Severe Grade |
|---|---|---|---|
| 1 | Score 1 | Score 2 | Score 3 |
| 2 | Score 1 | Score 2 | Score 3 |
| 3 | | Score 1 | Score 3 |
| Lung | | Score 1 | Score 2 |

Mild grade: 1 or 2 organs (but not lung) with maximum score of 1. Moderate grade: lung score 1 or ≥3 organs at score 1 or at least one organ at score 2. Severe grade: lung score 2 or score 3 in any organ.

In summary, we recommend applying the NIH 2024 criteria for organ severity scoring and global grading at the time of the initial diagnosis of cGvHD.

## 7. Current Strategies for Chronic GvHD Treatment

The treatment paradigm of cGvHD is in evolution, owing to a better understanding of its pathophysiology and the recent development of effective novel therapies. Systemic cGvHD therapy has three ideal goals, including (1) the induction of immunologic tolerance, (2) reversal and limiting organ damage and the preservation of affected organ function, and (3) the successful discontinuation of all systemic immunosuppression without the recurrence of GvHD and without relapse of hematological malignancy. When treating cGvHD, healthcare providers must be cognizant of the balance between the systemic immunosuppression required to control GvHD and the risk of infection/hematologic malignancy relapse and other complications associated with long-term immunosuppression.

Practical measures evaluating the efficacy of cGvHD treatment include: (1) the overall response rate based on the NIH proposed response criteria and a durable response after

the achievement of an initial response; (2) the clinical benefit (defined as complete/partial response, as well as a stable disease but a significant reduction or discontinuation of corticosteroids) [32]; (3) failure-free survival (FFS; defined as treatment switch due to an inadequate response/progression/intolerance to treatment, non-relapse mortality, or the recurrence of a hematologic malignancy) [33–35] and overall survival; (4) the duration of systemic immunosuppressive treatment with the rapid reduction and/or discontinuation of corticosteroids to avoid toxicities from prolonged corticosteroid exposure [36]; and (5) the regain of organ function, limit of adverse effects, and improvement in the quality of life (QoL), using the modified Lee Symptom Scale [6,36]. These measures have been frequently used in multiple clinical trials evaluating the efficacy of newer drugs for cGvHD treatment [13–15].

As shown in Figure 1, the choice of therapy is based on the affected organs or sites, the severity and extent of cGVHD, other comorbidities or medical issues, potential drug–drug interactions, logistics, and reimbursement/individual patients' drug coverage. Wide variability in practice is observed based on these factors and due to differences in institutional approaches and resources. The choice of steroid-sparing agent largely depends on the physician's experience and its biological mechanism of action on the immune system [10]. Children are unusually susceptible to the long-term side effects of steroid usage, including a much higher rate of osteonecrosis in puberty and the inhibition of bone growth leading to short stature as well as osteoporosis.

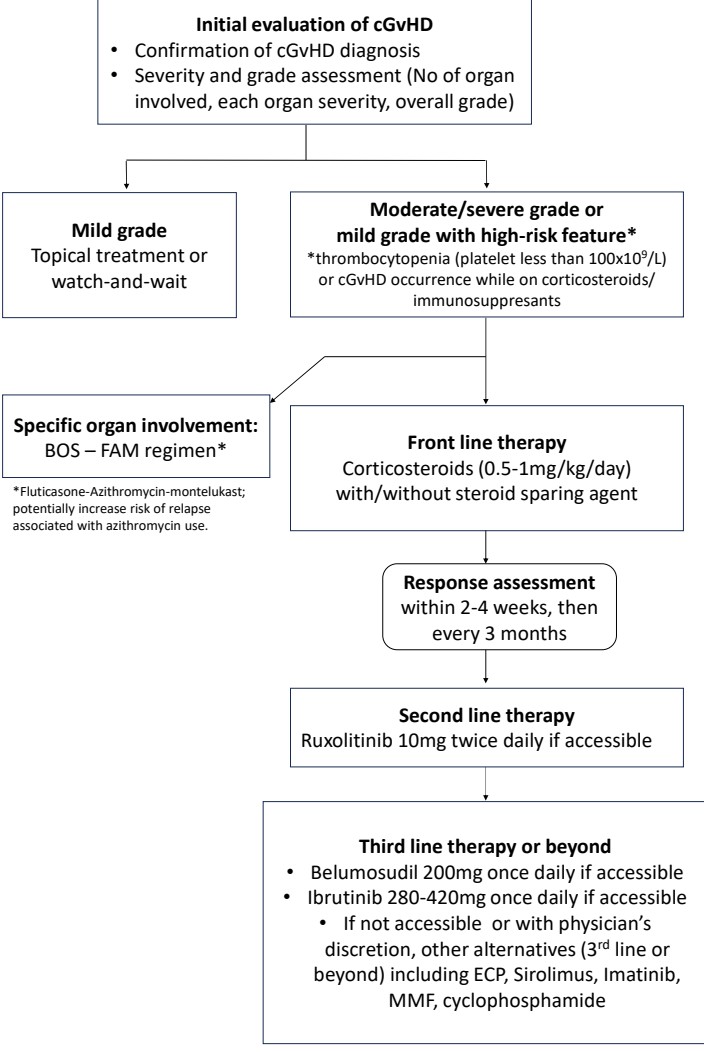

**Figure 1.** Proposed therapeutic algorithm for chronic GvHD treatment based on current knowledge.

In keeping with published guidelines, we recommend the following as our current standard practice of cGvHD treatment:

(1) Front-line therapy of moderate-to-severe grade chronic GvHD is corticosteroids (prednisone 0.5–1 mg/kg or equivalent) with or without the addition of another systemic agent for steroid-sparing purpose such as calcineurin inhibitors, as these agents were reported to alter the natural course of the disease [37,38].

(2) To avoid any unnecessary drug-related side effects or toxicities from systemic immunosuppression and to avoid drug–drug interaction; the number of systemic agents is advised to be minimized as much as possible. However, with advances in newer therapeutic agents, combination strategies may now be reconsidered.

(3) Local or topical therapy such as corticosteroid-containing cream, a steroid bronchodilator, or an enteric form of corticosteroids should be maximized to avoid systemic steroid-induced, long-term toxicities.

(4) Supportive management, antimicrobial prophylaxis, and access to physiotherapy or rehabilitation programs are strongly recommended to avoid long-term side-effects and toxicities from chronic exposure to corticosteroids.

(5) Regular assessment of the response to the treatment per target organ is advised to expedite corticosteroid tapering.

## 8. Current Front-Line Therapy for Chronic GvHD Treatment

Systemic therapy is indicated for patients with moderate-to-severe grade cGvHD, which is defined according to the NIH 2014 criteria as the involvement of three or more organs, moderate or severe grade organ involvement in any organ, or any highly morbid form of GvHD, such as lung or sclerotic GvHD [11]. Systemic treatment is also indicated for patients with mild grade cGvHD but presenting in conjunction with high-risk features such as thrombocytopenia (less than $100 \times 10^9$/L) or the development of cGvHD while on corticosteroid treatment. Symptomatic but mild grade cGvHD is often treated with topical therapies alone. Topical agents may also be used as adjuncts to systemic therapy to improve and accelerate the local response.

Front-line therapy for cGvHD treatment typically involves corticosteroids, which are potent anti-inflammatory agents that can suppress the underlying immune response causing cGvHD [11]. The initial dose of corticosteroids is based on the severity of the disease, with higher doses indicated for more severe cases. It is recommended that prednisone is to be started at the dose of 0.5 to 1 mg/kg/day, with the goal of achieving at least a partial response within 2–4 weeks. When the patient is responding to corticosteroids, the dose can be gradually tapered over several months, with careful monitoring for GvHD recurrence or steroid-related toxicity.

Additional drugs can be added for steroid-sparing purposes. Given that prolonged systemic corticosteroid treatment causes significant toxicity, including weight gain, bone loss, myopathy, diabetes, hypertension, anxiety/depression, cataracts, and increased risk of infection, the combined therapy of corticosteroids with a steroid-sparing agent is generally recommended. While a calcineurin inhibitor (CNI) is the most commonly used steroid-sparing agent, its inhibitory effect on Treg lymphocytes can be detrimental. Past history of intolerance to calcineurin inhibitors, renal function impairment, and previous microangiopathy needs to be carefully reviewed before its selection as the steroid-sparing agent [38].

Sirolimus (rapamycin) is a lipophilic macrocytic lactone with immunosuppressive properties [39]. Its unique properties give it potential advantages over other immunosuppressive agents, including (1) immunosuppressive action through T cell inhibition, while promoting CD4⁺CD25⁺FoxP3⁺ regulatory T cells (Tregs), a T cell population involved in graft-versus-leukemia reaction, (2) the inhibition of antigen presentation and dendritic cell maturation, (3) antifibrotic properties, (4) antineoplastic activity, (5) antiviral activity, and (6) the steroid-sparing effect. Although more frequently used for aGvHD prevention and treatment, its use as a steroid-sparing agent in cGvHD could be of interest [39,40].

Mycophenolate mofetil has failed to demonstrate its efficacy as a front-line therapy for cGvHD treatment, with an increased risk of hematologic malignancies and relapse [41], while azathioprine is also known to increase the risk of secondary malignancies [42]. Thus, several treatment guidelines, including the National Comprehensive Cancer Network (NCCN) guideline [18], do not recommend the use of mycophenolate mofetil for front-line therapy and have advised against the use of azathioprine for cGvHD treatment [43].

Steroid tapering is undertaken following the improvement or resolution of cGvHD-related symptoms and signs. Tapering schedules vary across different transplant centers [11]. While some taper on an alternate-day tapering schedule [11], others use a daily dose tapering policy. The efficacy of alternate-day vs. daily administration of corticosteroids has been reported in pediatric renal transplantation, but has not been tested in HCT [11].

In summary, we recommend systemic treatment in patients with moderate-to-severe grade cGvHD as well as mild grade cGvHD with high-risk features. We also recommend the use of systemic corticosteroids with consideration of steroid-sparing agents as the current standard treatment. Diverse options for steroid-sparing agents exist including CNI and sirolimus, which are widely available in Canada.

## 9. Response Assessment following Chronic GvHD Treatment

The NIH 2014 criteria provide provisional response criteria for evaluating the efficacy of therapeutic interventions in patients with cGvHD [44]. These criteria are based on the assessment of organ severity score, liver enzyme values, FEV1 value, photographic range of motion (P-ROM) score, or symptom score. Responses are classified as complete response (CR), partial response (PR), or lack of response (unchanged, mixed response, or progression), based on the improvement or worsening of the organ severity score, lab or PFT value, P-ROM score, or symptoms in each organ system. A detailed definition of each response, i.e., CR, PR, and lack of response, is summarized in Table 5.

**Table 5.** Summary of response assessment based on the NIH 2014 consensus criteria.

| Response | Definition |
| --- | --- |
| Complete response (CR) | (1) Resolution of all manifestations in each organ or site, and PR is defined as improvement in at least one organ or site without progression in any other organ or site. <br> (2) The skin, mouth, liver, upper and lower GI tract, esophagus, lung, eye, and joint/fascia are considered to evaluate the overall response. <br> (3) The genital tract and other manifestations are not included due to a lack of validated response measures. <br> (4) The CR category may not apply to organs with irreversible damage. |
| Partial response (PR) | An improvement in score from baseline that reflects a genuine clinical benefit and exceeds the measurement error of the assessment tool, as follows: an improvement of one or more points on a 4-to-7-point scale or an improvement of two or more points on a 10-to-12-point scale. |
| Disease progression (DP) | (5) For skin, eye, esophagus, and upper and lower GI tract, a worsening of one point or more on the 0 to 3 scale is considered progression, except a change from 0 to 1, which is considered trivial progression since it often reflects mild, non-specific, intermittent, self-limited symptoms and signs that do not warrant a change of therapy. <br> (6) For joint/fascia, a worsening of one point or more on the 0 to 3 scale is considered progression, even if from 0 to 1. For joints assessed using the P-ROM, a worsening of one or more points on the 7-point scales (wrist, elbow, or shoulder) and one or more points on the 4-point scale (ankle) is considered progression. <br> (7) Worsening of liver GvHD is defined by an increase of two or more times the upper limit of normal for the assay for alanine transaminase, alkaline phosphatase, or total bilirubin. <br> (8) For patients with lung involvement, absolute worsening of FEV1 by 10% predicted or more (e.g., 50% to 40%) is considered progression. |
| Mixed response | A new category defined as CR or PR in at least one organ accompanied by progression in another organ, while the cases that do not meet the criteria for CR, PR, DP, or mixed response are considered unchanged. |

Therefore, we strongly recommend evaluating the response to cGvHD treatment within at least 2–4 weeks of therapy starting, with regular assessment every three months.

### 10. The Current Definition of Corticosteroid Failure following Chronic GvHD Treatment

In cases where there is no response to initial therapy, or if the response is incomplete or transient, systemic treatment should be switched to another therapy. The definition of steroid failure following frontline treatment is not fully established for cGvHD, in contrast to steroid-refractory acute GvHD, which has been well defined and standardized. Per the NIH 2014 criteria, the following criteria were proposed for the diagnosis of steroid-refractory cGvHD [10,36]: (1) a lack of response or disease progression after the administration of minimum prednisone 1 mg/kg/day for at least 1 week (i.e., steroid refractoriness), (2) disease persistence without improvement despite continued treatment with prednisone at >0.5 mg/kg/day or 1 mg/kg/every other day for at least 4 weeks (i.e., steroid refractory cGvHD), and (3) an increase in the prednisone dose to >0.25 mg/kg/day after two unsuccessful attempts to taper prednisone (i.e., steroid-dependent cGvHD).

We recommend paying special attention to corticosteroid failure to determine steroid refractory cGvHD, based on the NIH 2014 criteria.

### 11. Current Consensus for Second-Line Therapy or beyond for Chronic GvHD after Steroid Failure

The choice of second-line therapy is typically ruxolitinib, based on a randomized controlled trial [13]. However, beyond second-line therapy, there is no standard and the treatment should be individualized based on the specific organ involvement, the severity and extent of cGvHD, the potential side effects of the medications, other comorbidities or medical issues, drug–drug interactions, logistics, and reimbursement. Before switching treatment due to progression or non-response in one organ, it is recommended that screening for other organ involvement by active cGvHD is undertaken by repeating PFTs and/or P-ROM score assessments.

Ruxolitinib, a Janus kinase (JAK) inhibitor, is the only drug that demonstrated a clinical benefit and superior efficacy over the best available therapy in a phase 3 trial as a second-line treatment [13]. The REACH3 study and Canadian real-world experience study have shown that ruxolitinib can lead to improvements in symptoms, overall response, and failure-free survival (FFS) and can reduce the need for corticosteroids [13,32,45]. Accordingly, it should be strongly considered as a second-line option for steroid-refractory cGvHD [33,45].

The choice of third-line or later therapies is not standardized and there are no well-controlled randomized trials to guide treatment selection. The following are variably available and effective cGvHD-active treatment options for third-line treatment or beyond: belumosudil [14], ibrutinib [46], ECP [8,9], sirolimus [8–11,44], mycophenolate mofetil, and rituximab. Table 6 summarizes the currently available and frequently used third-line and beyond treatment options in Canada [7]. Table 7 compares the three newest options, i.e., ruxolitinib, belumosudil, and ibrutinib, which had been approved by Health Canada.

(1) **Belumosudil:** Belumosudil is a selective inhibitor of rho-associated coiled-coil kinase 2 (ROCK2), which is involved in the signaling pathways that lead to inflammation and tissue damage in cGvHD. Belumosudil has been shown to be effective in the treatment of cGvHD, particularly in the cases of sclerotic or mild-to-moderate lung GvHD [14]. Clinical trials have shown that belumosudil can lead to improvements in symptoms, overall response, and FFS and can reduce the need for systemic immunosuppression [14].

(2) **Ibrutinib:** Ibrutinib is a Bruton's tyrosine kinase inhibitor that has been shown to be effective in the treatment of cGvHD. It works by blocking the activation of B cells and T cells, which play a key role in the development of cGvHD. Phase 2 clinical trials have shown that ibrutinib can lead to improvements in symptoms and organ function and can reduce the need for systemic immunosuppression [12]. However, a real-world, experience-based study reported only a 9% failure-free survival (FFS) rate at two years with a median of 4.5 months of FFS [46].

(3) **Extracorporeal photopheresis (ECP)**: ECP is a therapeutic apheresis procedure that entails the separation of activated T lymphocytes from the **patient's** blood, which

are subsequently subjected to ultraviolet light and a photosensitizing agent before being reinfused into the patient's circulation. ECP has demonstrated efficacy in the management of steroid-resistant cGvHD, particularly in the setting of sclerotic GvHD and in Canadian practice [8,9]. Several studies have demonstrated its corticosteroid-sparing effect and potential role in combination with other drugs. Accessibility, tolerability, and appropriate blood counts are pre-requisites to the successful use of this modality. Several months of ECP may be needed before tangible improvement can be seen.

(4) **Sirolimus**: This agent operates by suppressing the activity of mTOR, a protein involved in the regulation of cellular growth and proliferation. **Sirolimus** may be employed either as monotherapy or in conjunction with calcineurin inhibitors, and the dosage is tailored to the individual patient's response and tolerance [47]. Emerging data suggest sirolimus may be superior to cyclosporine or tacrolimus, based on its beneficial sparing effect on regulatory T cells [47,48], but this is still not confirmed in large trials with sufficient follow-up.

(5) **Other systemic agents**: Other immunosuppressive agents, such as mycophenolate mofetil [49], rituximab, imatinib [50], methotrexate, or **cyclophosphamide**, may be used as salvage therapy in cases of steroid-refractory cGvHD. These medications are often reserved for cases where other therapies have failed, as they can be associated with significant toxicity. Axatilimab is also very promising and is awaiting approval by the US-FDA, as of December 2023 [15].

Multiple aspects need to be considered for third-line therapy or beyond such as the mode of action of the drug, organ-specific treatment outcome, infection history, comorbidity, relapse risk, compliance/logistics, and funding and treatment access. The organ-specific action of the treatment should also be taken into consideration. For instance, cyclophosphamide has been adopted for renal GvHD manifesting as nephrotic syndrome [51], while the FAM regimen (i.e., fluticasone-azithromycin-montelukast) has been used for pulmonary GvHD [52,53], although there is a concern for an increased risk of relapse associated with the use of azithromycin [54]. It is recognized that ruxolitinib has little evidence supporting its use in some of the atypical cGvHD manifestations (e.g., immune cytopenias and renal and neurologic manifestations) and that other immunosuppressive agents (e.g., rituximab) may be preferable in these settings.

In conclusion, we recommend ruxolitinib as a second-line therapy in the patients experiencing corticosteroid failure following front-line cGvHD treatment, while other factors are to be considered for other therapeutic options. As a third-line option or beyond, newer agents, other systemic agents, and ECP can be used, while considering their funding accessibility.

**Table 6.** Summary of commonly used treatment options in Canada for steroid-refractory chronic GvHD as second-line therapy or beyond, reprinted/adapted from Ref. [10].

| Therapy | Type | Recommendation | Overall Response | Overall Survival | Toxicities | Study Type |
|---|---|---|---|---|---|---|
| Ruxolitinib | Janus kinase 1/2 inhibitor | ≥second-line | BOR 76% (CR 12%, PR 64%) in 165 patients with SR-cGvHD [13]; 85% (CR 7%, PR 78%) in 41 patients with SR-cGvHD [33,46,55] | 97% at 6 months [55] | Viral reactivation/infection, peripheral neuropathy, anemia, thrombocytopenia, and neutropenia [13,56]; viral reactivation, cytopenia, and malignancy relapse [55] | Phase 3 randomized trial |
| Ibrutinib | Bruton's tyrosine kinase inhibitor | ≥third-line | BOR 67% (CR 21%, PR 45%) in 42 patients with cGvHD, with median follow-up of 13.9 months [12] | 71% at 2 years in cGvHD [57] | Pneumonia and impaired platelet function [58] | Phase 2a trial |

**Table 6.** *Cont.*

| Therapy | Type | Recommendation | Overall Response | Overall Survival | Toxicities | Study Type |
|---|---|---|---|---|---|---|
| Extracorporeal photopheresis | UVA treatment of mononucleated blood cells via leukapheresis | ≥second-line | Rates dependent on site and severity—highest responses in skin, liver, mouth, and BOS [59–62]: 67% (CR 23%, PR 44%) in 48 patients with SR-cGvHD [61] | 53–78% at 1 year [8,9,11,59]. | Vascular access complications [58] | Phase 2 randomized trial |
| Mycophenolate mofetil | Antimetabolite immunosuppressant | ≥third-line | 26–64% [11,49] | 67–96% at 1 year [11] | Viral reactivation, hypertension, pneumonia, and post-transplantation lymphoproliferative disease [58] | Retrospective cohorts |
| Rituximab | CD20 (B cell surface antigen) monoclonal antibody | ≥third-line | 65% in 38 patients with SR-cGvHD [59]; 70% (CR 10%) in 20 patients with SR-cGvHD [63]; 27% in 37 patients with sclerotic cGvHD [64]; 17% (CR 17%) in 6 patients with SR-cGvHD [64] | 72% at 1 year; 76% at 2 years [11,55] | Infections, infusion-related symptoms, and late neutropenia [62,63] | Phase 2b randomized trial |
| Sirolimus | mTOR inhibitor | ≥third-line | 81% (CR 38%, PR 43%) in 47 patients with SR-cGvHD [65]; 94% of 16 patients with cGvHD [47,66] | – | Thrombotic microangiopathy, renal insufficiency, and proteinuria [65–67] | Phase 2a trials |
| Imatinib | Multi-kinase inhibitor | ≥third-line | 79% (CR 37%, PR 42%) in 19 patients with SR-cGvHD [68]; 26% in 35 patients with sclerotic cGvHD [64] | 84% at 1.5 years [68] | Fluid retention, myelosuppression, and anemia [68] | Phase 2b trial |
| Cyclophosphamide (either pulse or low dose) | Alkylating agent | ≥third-line | 100% of 3 patients with cGvHD showed response in treatment of skin and oral cavity [69]; 60% of 15 patients showed improvement after 8–12 monthly cycles [70] | – | Short-term myelosuppression, neutropenia, fatigue, and nausea [51,69,70] | Retrospective cohorts |
| Belumosudil | ROCK2 inhibitor | ≥third-line | 74% (CR 3%, PR 71%) of 132 patients with cGvHD [14] | FFS 77% at 6 months [14] | Pneumonia, hypertension, hyperglycemia, and increased gamma-glutamyltransferase [14] | Phase 2 open-label, randomized clinical trial |
| Axatilimab | IgG4 antibody targeting the CSF-1 receptor | Available in clinical trial only | 58% of 12 patients with cGvHD across doses [15] | – | Increased gamma-glutamyltransferase, asparatate aminotransferase, and creating phosphokinase, periorbital edema [15] | Phase 1/2 dose-escalation and dose-expansion study |

Adopted and modified from Bone Marrow Transplantation **2021**, *56*, 2079–2087 [10]. Abbreviations: BOR; best overall response; cGvHD, chronic graft-versus-host disease; CR, complete response; PR, partial response; BOS, bronchiolitis obliterans; UVA, ultraviolet ray-A; SR-cGvHD steroid-refractory chronic GvHD; FFS, failure-free survival.

**Table 7.** Summary of three newer treatment options for steroid-refractory chronic GvHD approved by Health Canada.

|  | Ibrutinib | Belumosudil | Ruxolitinib |
|---|---|---|---|
| Publication | Miklos (2017) [12], Phase 1b/2 | Cutler (2021) [14], Phase 2 | Zeiser (2021) [13], Phase 3 |
| No. of patients | $n = 42$ | $n = 66/66$ | $n = 165$ |
| Dose | 420 mg once daily | 200 once daily/200 mg twice daily | 10 mg twice daily |
| Indication | Second-line or beyond | Third-line or beyond | Second-line or beyond |
| Follow-up/exposure | 13.9 months | 12 months | 9.5 months |
| ORR at 6 months | NR | NR | 50% (CR 6.7%) |
| ORR, max | 67% (CR 21%, PR 45%) | 74–77% (CR, 5.3%) | 76% (CR 12.1%) |
| Steroid stop | 5/42 at 12 months (12%) | 21% | - |
| FFS duration | NR | 14–15 months | Not reached |
| FFS at 12 months | NR | 56% | 62% |
| Adverse events | Fatigue, diarrhea, muscle spasms, nausea, and bruising | Fatigue, diarrhea, nausea, and upper respiratory tract infection | Anemia and thrombocytopenia |
| Public reimbursement | Not available | Under review (third-line or beyond) | Available (second-line or beyond) |

Abbreviations: ORR, overall response; NR, not reported; CR, complete response; PR, partial response; FFS, failure-free survival.

## 12. Future Directions for Chronic GvHD Treatment Development

The NIH 2020 consensus report highlighted the importance of the early clinical recognition of cGvHD, even before meeting NIH diagnostic criteria [20], early biomarker development for cGvHD diagnosis, and early diagnosis when presented in the form of atypical GvHD [22]. It also underscores the fact that when patients meet the NIH 2014 criteria, many of them already have a high burden of disease and fibrotic insults to their organs [21]. Accordingly, pre-emptive therapy for future development was emphasized [23].

With recent advances in cGvHD therapeutics and the approvals of ibrutinib [12], ruxolitinib [13], belumosudil [14], and the soon-to-be approved axatilimab [15], the therapeutic paradigm is rapidly evolving. Here is a summary of its developmental direction.

1.  **Mode-of-action-based cGvHD drug selection:** An emerging concept of cGvHD treatment involves selecting therapies based on their modes of action. As discussed, the pathogenesis of cGvHD can be stratified into inflammatory, immune deregulation, and fibrotic phases [10]. Each drug has its own mode of action that targets one or more of these phases. For example, ruxolitinib has robust anti-inflammatory activity, while belumosudil and axatilimab have strong antifibrotic properties. Depending on the clinical manifestation and suspected pathogenesis of cGvHD in a particular patient, the most appropriate could be selected.

2.  **Combination strategy:** While minimizing the number of immunosuppressive drugs was one of the main principles of cGvHD management in the past, with the introduction of newer targeted drugs, a combination strategy is now rational and revisited. One such combination strategy could include a novel agent, e.g., ruxolitinib or belumosudil, along with another therapeutic agent that has been extensively evaluated [71]. A combined approach of ECP with belumosudil or with ruxolitinib seems promising and is currently under investigation [72]. Alternatively, a combination of two novel agents is also promising and might be therapeutically valuable, if accessible.

3.  **Steroid-free regimen:** Development of a corticosteroid-free, front-line regimen is highly warranted. Corticosteroids have been the mainstay of cGvHD treatment for

over five decades. When feasible, clinical trials of initial systemic therapy should investigate steroid-free therapeutic strategies in front-line therapy [71].

4. **Pre-emptive approach:** In order to improve the quality of life and long-term outcomes in patients suffering from highly morbid forms of cGvHD, a pre-emptive treatment approach could be an optimal approach. This strategy could potentially prevent the progression of cGvHD in high-risk patients destined to develop severe and highly morbid forms of cGvHD [23]. Such an approach will likely require further research on cGvHD biomarker development.

## 13. Multidisciplinary Approach and Supportive Care in the Management of Chronic GvHD

The management of cGvHD requires a multidisciplinary approach, as it involves the coordination of multiple medical specialists to address the various symptoms and complications associated with cGvHD.

The multidisciplinary team typically includes specialists in infectious diseases, ophthalmology, dermatology, gynecology, gastroenterology, respirology, hepatology, nephrology, dentistry, psychiatry, and rehabilitation medicine. In addition to specialist physicians, the multidisciplinary team must include pharmacists, nurse practitioners, physician assistants, dietitians, physiotherapists, and social workers. A dedicated team approach with relevant expertise is necessary to provide optimal and holistic care for patients dealing with a chronic illness that can significantly affect their QoL. Also, the development of a dedicated "cGvHD program" in each center that operates in parallel with a dedicated multidisciplinary team has the potential to improve the quality of care, patient-reported outcomes, and QoL in cGvHD patients.

Chronic GvHD is associated with a severe impairment in QoL [73]. Patients with cGvHD may present with a diverse range of manifestations affecting multiple organ systems. These symptoms may be intrusive and can include chronic pain, fatigue, xerostomia and xerophthalmia, skin rashes and ulcers, joint stiffness, dyspnea, and intractable and persistent gastrointestinal disturbances such as chronic diarrhea and nausea. Supportive measures to control and improve these symptoms are needed and may include physiotherapy, exercise, pain clinic consultation, and the care of many specialists.

Patients with cGvHD experience excess comorbidities versus survivors of allogeneic HCT without cGvHD, including loss of bone density, hyperlipidemia, hyperglycemia, and subsequent malignancies amongst others. Consideration should be given to regular bone mineral densitometry, measurement of glucose and lipids, and age-appropriate malignancy screening [74]. In addition to the physical manifestations, cGvHD can exert a considerable financial, emotional, and psychological toll on patients. The disease can be prolonged, necessitating ongoing medical intervention and monitoring, which may trigger feelings of anxiety, depression, and social isolation. The burden of disease management may also impact patients' ability to carry out daily activities, work, and interact with others, further impinging upon their quality of life.

A multidisciplinary approach to the management of cGvHD strives to address these diverse aspects of the disease and to enhance patients' QoL. This may involve treating the physical symptoms with pharmaceutical and supportive modalities, providing psychological and social support, and facilitating patients' navigation of the practical aspects of their treatment, such as medication adherence and appointment scheduling. By attending to these varied needs, the multidisciplinary team can assist patients with cGvHD in sustaining their independence and promoting their overall well-being.

**Supplementary Materials:** The following supporting information can be downloaded at: https://www.mdpi.com/article/10.3390/curroncol31030108/s1, Table S1. Summary of Chronic GvHD systematic review [11].

**Author Contributions:** Conceptualization, D.D.H.K. and K.R.S.; writing—original draft preparation, D.D.H.K.; writing—review and editing, D.D.H.K., G.P., K.L., K.P., D.A., R.V.N., S.L., U.D., J.W., M.E., K.J., C.F., C.L., I.N.-B., A.D.L., R.K., I.W. and K.R.S.; visualization, D.D.H.K.; supervision, D.D.H.K.,

I.W., and K.R.S.; project administration, D.D.H.K. and K.R.S. All authors have read and agreed to the published version of the manuscript.

**Funding:** This research received no external funding.

**Institutional Review Board Statement:** Not applicable.

**Informed Consent Statement:** Not applicable.

**Data Availability Statement:** Not applicable.

**Acknowledgments:** The current work was conducted without financial support from pharmaceutical companies.

**Conflicts of Interest:** D.D.H.K.—honoraria from Novartis, Sanofi, mallinckrodt and Jazz, and research grant from Novartis and Sanofi. Advisory board member and constancy for Novartis and Sanofi; G.P.—honoraria and/or consulting: sanofi, servier, mallinckrodt, abbvie, jazz, medexus, pfizer, seattle genetics, takeda, amgen, merck, gilead, novartis, kyowa kirin, paladin, sobi / research support: Syndax, Abbvie, Equilium—for being local PI; K.L.—honorarium from Sanofi and advisory boards for Novartis and Sanofi; K.P.—advisory board for Sanofi; J.W.—honoraria from Sanofi and Novartis; M.E.—honoraria from Novartis, Sanofi and Advisory board member and constancy for Novartis and Sanofi; K.J.—Ad board: Sanofi, Paladin, Jazz, Avir, Pfizer, Vertex/Research funding: Jazz; I.W.—Research funds and Advisory Board-Sanofi; K.R.S.—Ad Board: Novartis, Sanofi, Incyte and DSMB: BMS, Seres. Other authors declare no conflicts of interest.

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
