# Peer review of "Cell Therapy Transplant Canada (CTTC) Consensus-Based Guideline 2024 for Management and Treatment of Chronic Graft-Versus-Host Disease and Future Directions for Development"

_curroncol, doi:10.3390/curroncol31030108_

Round 1

Reviewer 1 Report

Comments and Suggestions for Authors

This is a consensus-based Canadian guideline whose primary purpose is to standardize and facilitate the management of cGVHD across Canada. It is very well written, appropriately detailed and serves an important mission.

I just had two minor questions:

1. Table 4 doesn't list some agents that are in the NCCN guidelines and used in the US (such as abatacept or low dose IL-2). Is this because they are not used in Canada? 

2. The authors recommend regular assessments every 3 months after the first 2-4 weeks. Isn't that treatment interval too long for patients who are on tapering doses of steroids and tend to develop multiple complications?

Author Response

Dear Reviewer

Thanks for your valuable comments. It really helps to improve our manuscript.

Followings are my answer to your inquiries.

1. Table 4 doesn't list some agents that are in the NCCN guidelines and used in the US (such as abatacept or low dose IL-2). Is this because they are not used in Canada?

Answer: Thanks for pointing it out. Yes, correct. Group consensus is to remove abatacept or low dose IL-2 which are not currently available and not used in Canada.

2. The authors recommend regular assessments every 3 months after the first 2-4 weeks. Isn't that treatment interval too long for patients who are on tapering doses of steroids and tend to develop multiple complications?

Answer: Thanks again to point this issue out. In some centers, the patient will be monitored more frequently by community hematologist while transplant center monitor them less frequently. Accordingly, our statement is a minimum interval to monitor their response. At least, the patient's response needs to be assessed within first 2-4 weeks particularly for drug-related toxicity, followed by regular assessment at least every 3 months. I hope it is sufficient to explain our statement.

Reviewer 2 Report

Comments and Suggestions for Authors

Authors created this paper to standardize and facilitate the management of cGvHD across Canada. I believe the content in this paper is based on the NIH consensus guideline and is acceptable.

However, I feel that this paper is a comprehensive review of cGvHD and lacks enough information as a clinical guideline for readers in Canada.

As shown in Figure 1, the initial evaluation of cGvHD is very important. Since clinical symptoms are described using only words and no figures or pictures in this paper, readers may have difficulty diagnosing cGvHD. The severity score for each organ is not fully described in Table 3A. I think more information should be added.

The judgment of changing from 2nd line treatment to 3rd line is very interesting and important. I would like to know the authors' opinion and if possible, add it to this paper.

In lies 52-53, what do the underlines mean in this sentence?

Author Response

Dear Reviewer 2,

I really appreciate your thoughtful advice and comments on our draft. It is very valuable to improve the quality of our manuscript.

Followings are the answers for your comment.

  1. As shown in Figure 1, the initial evaluation of cGvHD is very important. Since clinical symptoms are described using only words and no figures or pictures in this paper, readers may have difficulty diagnosing cGvHD. The severity score for each organ is not fully described in Table 3A. I think more information should be added.

Answer: Thanks for this comment. As suggested and encouraged, we will add the summary of cGvHD severity scoring system in each organ as Table 3 while move out other tables.

  1. The judgment of changing from 2nd line treatment to 3rd line is very interesting and important. I would like to know the authors' opinion and if possible, add it to this paper.

Answer: Thanks for pointing this out. Yes, it is not well in agreement which treatment should be prioritized over others as 3rd line treatment option. We have to admit that it is very difficult to conclude with a single simple statement.

Response: As advised, following sentence was added before the paragraph for “organ specific action”.

“Multiple aspects need to be considered for 3rd line therapy or beyond such as mode of action of the drug, organ-specific treatment outcome, infection history, comorbidity, relapse risk, compliance/logistic and funding and treatment access.”

  1. In lies 52-53, what do the underlines mean in this sentence?

Answer: During the submission, it was accidentally underlined. I removed it.

Round 2

Reviewer 1 Report

Comments and Suggestions for Authors

Accept in present form

Reviewer 2 Report

Comments and Suggestions for Authors

I appreciate that the author has effectively revised the manuscript to meet my requirements. I believe that clinicians undergoing allogeneic hematopoietic stem cell transplantation in Canada will gladly find this paper useful.